# A Rare Skeletal Disorder, Fibrous Dysplasia: A Review of Its Pathogenesis and Therapeutic Prospects

**DOI:** 10.3390/ijms242115591

**Published:** 2023-10-26

**Authors:** Ha-Young Kim, Jung-Hee Shim, Chan-Yeong Heo

**Affiliations:** 1Interdisciplinary Program in Bioengineering, Seoul National University, Seoul 08826, Republic of Korea; hkim247@snu.ac.kr; 2Department of Plastic and Reconstructive Surgery, College of Medicine, Seoul National University, Seoul 03080, Republic of Korea; 3Department of Plastic and Reconstructive Surgery, Seoul National University Bundang Hospital, Seongnam 13620, Republic of Korea; xmylife@empas.com; 4Department of Research Administration Team, Seoul National University Bundang Hospital, Seongnam 13620, Republic of Korea

**Keywords:** fibrous dysplasia, bone disorder, GNAS mutation, variant G protein, rare disease, pathogenesis

## Abstract

Fibrous dysplasia (FD) is a rare, non-hereditary skeletal disorder characterized by its chronic course of non-neoplastic fibrous tissue buildup in place of healthy bone. A myriad of factors have been associated with its onset and progression. Perturbation of cell–cell signaling networks and response outputs leading to disrupted building blocks, incoherent multi-level organization, and loss of rigid structural motifs in mineralized tissues are factors that have been identified to participate in FD induction. In more recent years, novel insights into the unique biology of FD are transforming our understandings of its pathology, natural discourse of the disease, and treatment prospects. Herein, we built upon existing knowledge with recent findings to review clinical, etiologic, and histological features of FD and discussed known and potential mechanisms underlying FD manifestations. Subsequently, we ended on a note of optimism by highlighting emerging therapeutic approaches aimed at either halting or ameliorating disease progression.

## 1. Introduction

Fibrous dysplasia (FD) is a rare benign skeletal disorder that presents with a unique non-hereditary and idiopathic nature, in which fibro-osseous tissue replaces the normal bone [1]. In acute cases, it can lead to physical debilitation and a consequently high susceptibility to fractures and breaks. Although data estimate the incidence rate of FD worldwide to be 1 in 5000 to 10,000 cases, the accurate measurement of occurrences is rather obscure due to the asymptomatic clinical manifestation often presented in FD patients [2]. The number of FD cases is generally speculated to constitute 2.5–5% of all benign bone lesions [3].

Given the rarity of the disorder and likelihood for discovery, there are only a handful of cases in the literature to date that reflect this disease’s pathogenesis. However, with increasing findings being made about FD, the enigma behind the physiological and pathological aspects are becoming unraveled. In this review, we provided insights on the clinical, etiologic, and histological features of FD and examined current and potential pathogenic mechanisms involved in FD development. However, the current lack of valid models and treatments for tackling this disease brings to light the challenges in recapitulating the presentation of this disease in cellular and animal systems for an adequate translation of preclinical findings. In view of our interpretations on the pathophysiologic nature of the disease, we advanced our working hypothesis that FD exhibits unique biologic features in reflection of the complex, multifaceted process to dysplasia that can be effectively treated with a precise selection of targets.

## 2. Fibrous Dysplasia: Clinical Heterogeneity

Affecting a small number of individuals, predominantly children and young adults with neither recognized gender predilection nor hereditary components, it can occur sporadically in any portion of the skeleton, with no identifiable clinical pattern in growth distribution. Based on its clinical presentation, FD can be categorized into three subtypes: monostotic fibrous dysplasia, polyostotic fibrous dysplasia, and McCune–Albright syndrome (MAS). Considerably the most prevalent at 80% of FD cases is monostatic fibrous dysplasia, which is attributed to an isolated occurrence involving a single skeletal site, contrary to the multiple affected sites associated with polyostotic fibrous dysplasia [4]. Limited to a minority of polyostotic FD patients, approximately 3% of them develop MAS, a rare form of polyostotic FD that has been reported to be variably linked to cutaneous and endocrine functional disturbances in the clinical form of café-au-lait cutaneous hyperpigmented macules and hyperfunctioning endocrinopathies, including hyperthyroidism, growth hormone excess, overstimulated gonadal function, and cortisol excess [5].

The epidemiology of FD remains to be poorly understood not only due to its rarity in occurrence, but also because mild, asymptomatic FD often goes undiagnosed. Depending on the site of the lesion, symptoms also vary significantly. Despite the erratic nature of FD, it has a predilection for the long bones of the legs, arms, ribs, and spine and craniofacial bones. If present in the long bones of the limbs, it may be asymptomatic or present with mild-to-moderate pain, local swelling, and visible curvature and/or deformity of the bone. Lesions affecting the spine may exhibit a more conspicuous aberrant lateral curvature, resulting in scoliosis, and those present in the craniofacial region may engender diverse symptoms associated with pain, nasal congestion, uneven jaws, bulging eyes, and facial asymmetry [6]. Its pervasion in craniofacial spaces can compress adjacent nerves to generate the detrimental consequences of vision loss and hearing impairment [7]. In extreme cases, aggressive forms of FD can even degenerate into malignant tumors, osteosarcoma, and chondrosarcoma [8]. Although a spontaneous regression of lesions rarely, if ever, occurs, it is more likely for non-aggressive forms to remain quiescent without discernible growth, which is often observed in patients past the age of 30 [9]. Together, these clinical findings indicate that a FD diagnosis is relatively spared late in the disease development due to its rather few, if any, symptoms during the early stages of the disease process, which, in consequence, may require more invasive forms of treatment than non-invasive, conservative approaches.

## 3. Somatic Mosaicism and Its Clinical Relevance

FD has been linked to a mutation in a principal bone-forming gene known as GNAS, the guanine nucleotide-binding, alpha stimulating complex locus, which is situated in chromosome 20q13.3 that encodes the alpha subunit of the heterotrimeric Gs protein. It is assumed to arise from postzygotic de novo mutational events to bear the status of somatic mosaicism and contribute to clinical heterogeneity [10]. Despite recent animal studies suggesting possible heritability of the GNAS mutation via germline transmission, a lack of documented inheritance of the disease in humans may imply embryonic lethality in the case of germline mutation occurrence [11,12,13]. Based on this accord, FD has been regarded a non-hereditary condition that is not expressed via germline mutations but via direct focal expression of somatic mutations in local cells at the affected site [14]. Consistent with the proposed mutational origin, the GNAS mutation is not ubiquitously expressed in all tissues and cells, even in resident cell populations of the local disease-affected tissue. Interestingly, a heterogenous population comprised of mutants and wild-type bone mesenchymal stem cells (BMSCs) were discovered within the lesion [15]. Similarly, an in vivo evaluation of single culture and co-culture models using mutants and wild-type BMSCs indicated a successful reproduction of FD-like lesions only under co-culture conditions, which demonstrates the cellular requirements that are essential for robust cell growth and viability [16,17]. However, over time, it appears that these mutant cells are negatively selected to decrease in number for the gradual cessation of the lesion’s growth [18]. This phenomenon explains the age-related changes in the tumor progression rate and the eventual stabilization of FD observed typically after adolescence.

The question, however, remains about the highly diverse and variable phenotypic manifestations presented by FD. As mentioned earlier, it can manifest as an isolated monostotic lesion or polyostotic lesions with multiple organ involvement. Such clinical discrepancies are attributable to differences in the relative timing of occurrence of the GNAS mutation, whether it occurred during embryonic development or during the early postnatal periods [5]. Mutations introduced in progressively earlier times of the embryonic developmental phase are expected to have extensive skeletal involvement, depending on the germ layer affected. It can result in mutation expression in various cell types, such as osteoblasts, endocrine cells, and melanocytes, to subsequently display abnormal phenotypes characteristic of MAS involving bone lesions, endocrine dysfunctions, and café-au-lait pigmentation spots [10]. Contrarily, mutation exposure at later times of fetal development will yield a lower number of mutated cells, but its dispersion will contribute to the development of multiple bone lesion formations, identifiable as polyostotic FD. If occurred postnatally, mutation expression in the more committed cell lineage progenies lead to the confined, localized progression of FD, known as monostotic FD. The early and late designations of mutation incidence is, thus, critical in discerning the etiology of this disorder. However, the difficulty in accurately detecting low-frequency variants in the extremely heterogenous nature affiliated with this disorder and the challenge to clinically diagnose it from the broad spectrum of age-dependent phenotypic variation, as seen in patients, pose a major hurdle for clinicians to achieve this objective.

FD patients have exhibited high incidence levels of two main mutations within exon 8 of the GNAS gene. The majority of FD cases harbored a specific point mutation in codon 201, with the most common being the replacement of an arginine with a histidine, R201H, at a frequency of 80.5%, followed by replacement with a cysteine, R201C, 19,5%, and in rare cases, replacement with a serine, R201S, leucine, R201L, glycine, R201G, or proline, R201P [19,20,21,22,23,24] (Table 1). Seldom, patients exhibited mutations in codon 227 of exon 9, which involved the substitution of a glutamine with a lysine, Q227K, a leucine, Q227L, an arginine, Q227R, or a histidine, Q227H [25]. Approximately half of all patient cases have presented a mutated GNAS gene, but the remainder were not linked to any type of known mutation associated with FD. For some time, various studies have suggested the crucial role of the GNAS mutation in the disease development of FD. In most cases that demonstrate an absence of mutation involvement, its etiologic process remains to be unknown and the underlying pathologic mechanism needs to be elucidated.

It is, nonetheless, important to rule in the possibility of a missed diagnosis due to relatively low levels of somatic variants found in biopsied tissues of patients, i.e., the blood, skin, and bone. Variable levels of mutants were discovered depending on the tissue tested; in the context of blood samples of patients, digital PCR analysis achieved a detection rate of up to 37.8%, which had earlier tested negative via Sanger sequencing [26]. This suggests the need to establish more reliable diagnostic and molecular profiling techniques for detecting GNAS-related genetic variants in FD patients to accurately distinguish between true- and false-negative results. With respect to samples showing an absence of mutations, despite possible technical interference, given the heterogenous genomic nature of FD, we cannot exclude the possibility of novel variants for cases with a missing genetic diagnosis.

## 4. Histopathology and Cytopathology

The fibro-osseous anomaly in FD disrupts bone maturation and development to harbor disordered and incoherent architectural features and compromised integrity of considerable low mineral density. Normally, the bone remodeling process entails the replacement of primitive woven bone with mature lamellar bone characterized by highly organized layers of collagen fibers, and depending on the fate of the lamellae into osteons or trabeculae packets, either a compact cortical bone or a spongy bone is formed [27]. The degree of abnormal bone architecture and mineralization within a FD lesion has been revealed to be determined via the amount of layering; a few layers yield woven bone, whereas well-formed layers form lamellar bone [28]. Even though distinct layering can be found in fibrous dysplastic bone, its bone architecture can never be likened to the normal appearance of the lamellar structure found in a cortical bone.

Histological analyses of FD lesions suggest de novo bone formation in the form of “spongified” or cancellous bony structures embedded in the marrow stroma pervaded via an unmineralized, collagenous matrix [29]. The paucity of bony trabeculae interspersed within the fibrotic marrow spaces gives it an appearance of “Chinese characters” or an “alphabet soup” of thin and curvy lines. A closer examination of the bony trabeculae shows an abnormal formation of thick osteoid that is a clear indication of faulty mineralization [30]. In most cases, there is an absence of osteoblastic rimming along the borders; however, a heterogenous population of mutant and wild-type skeletal progenitor cells has been found residing within the exterior and the ectopic interior of the trabeculae alongside many TRAP-positive osteoclasts [28]. Large areas of the bony trabeculae, with major exposure to the connective tissue stroma, is particularly evident of osteolytic activities as opposed to smaller areas of the bony trabeculae that present with very early stages of bone formation [31]. The hypercellular stroma appears to consist of multiple cell populations: “fibroblastic” stellate-shaped stromal cells, few abnormal/normal osteoblasts, and osteocytes [32,33], as well as immune cells, such as myeloid precursor cells, osteoclast-like giant cells, macrophages, dendritic cells, and naïve and memory T cells [34].

Such observations suggest repeated episodes of impaired bone growth and excess matrix production by mutated osteoblasts amidst unrestrained osteoclastogenesis brought on via the hyperstimulated activities of the osteoclasts. The resulting disruption of bone formation and resorption balance is believed to be primarily responsible for the histopathologic pattern of trabecular remnants scattered in the dysplastic fibrous stroma [18,35]. Although the nature of FD’s pathogenesis remains to be further clarified, a growing body of evidence points to the intricacy and complexity behind the process that implicates non-linear, multi-level interactions between various cell types in the presence of multiple extrinsic cues, i.e., cell fate signaling and cell–cell interactions, and intrinsic regulations, i.e., epigenetic factors and transcription factors [36]. From this perspective, analyses of key target genes and aberrant signaling crosstalk among cell populations residing in and around the affected site is of major relevance to better understand the mechanistic details for disease development.

## 5. Pathogenesis

As pathology practice continues to advance, novel mechanisms underpinning critical pathways for rare disease development are being discovered to unearth putative targets for therapeutic engagement. Our understanding of FD has also evolved over the years to reveal unprecedented insights into the biology and pathophysiology of FD. Nonetheless, despite significant progress to capture FD development, these analyses are often inherently one-sided in its interpretation, providing a perspective on individual mechanistic processes. As aforementioned, a holistic approach is necessary to fully appreciate the dynamics and intricacies of bone biology and metabolism. In the following section, we discuss current and potential pathogenic mechanisms underlying classical FD manifestations involving aberrant bone formation, fibrous matrix development, bone remodeling imbalance, and bone pain.

### 5.1. Aberrant Bone Formation

Bone formation can be largely divided into three steps: proliferation and differentiation, maturation, and mineralization. It is a process governed via local environmental factors, such as growth factors, cytokines, hormones, and mechanical stimuli [37,38]. Its initiation begins with the activation of two primary pathways, namely the Wingless/Int-1 (Wnt)/β-catenin and transforming growth factor-β (TGF-β)/bone morphogenic protein (BMP) pathways, in response to humoral and/or mechanical signals that cause mesenchymal stem cells (MSCs) residing in the bone marrow stroma to commit and differentiate into early stages of the osteoblast lineage [39]. In turn, a series of transcription factors, i.e., runt-related transcription factor 2 (RUNX2) and osterix (OSX), and associated bone matrix proteins, i.e., type I collagen (COL1), alkaline phosphatase (ALP), and non-collagenous proteins, are expressed to promote the maturation of pre-osteoblasts [40,41,42].

The final and most essential step of bone formation, the mineralization process, is implemented by mature osteoblasts. These cells secrete large amounts of collagen type 1 for matrix deposition and ALP for extracellular matrix (ECM) maturation, followed by the expression of osteopontin (OPN) for apatite growth, osteocalcin (OC) for apatite alignment with the collagen fibril axis, and bone sialoprotein (BSP) for matrix mineralization [43]. As evidenced by their roles in bone formation, the extent and character of new bone formation is not only dictated by the osteogenic capacity of osteoblasts as a single unit, but also as a colony, representative of the number and the life span of mature osteoblasts [44]. Another major constituent of the basic multicellular unit (BMU) of the bone cells is the flat bone-lining cells of the osteoblast lineage that modulate bone resorption via prevention of the osteoclast interaction by sealing off inactive bone surfaces [45]. The accumulation of pre-mineralized matrix takes the form of osteoid. In the final stages, osteocytes, which are terminally differentiated cells of the osteoblast lineage, become embedded in the bone matrix [46]. These different cellular components are required to act in a coherent and synchronous manner for the micromanagement of bone health, and any divergence in cellular activity can alter the biological functions for abnormal bone formation. In FD, the skeletal aberrancies that are encountered are speculated to originate from a single point mutation of the GNAS gene in BMSCs that invariably alter signaling pathways associated with mineral and bone development to lead to faulty bone mineralization. For this reason, the impact of this mutation on the osteoblast differentiation process was closely examined by determining changes in the osteoblast activity-associated key marker genes (Figure 1).

#### 5.1.1. Enhanced Proliferation and Osteoblast Differentiation

Previous studies on FD have demonstrated significantly elevated cAMP levels induced via a sustained activation of the G protein α subunit, Gsα, and adenylate cyclase in mutated BMSCs [49,50,51]. Considering that the cAMP signaling pathway is a key intracellular signaling pathway involved in a wide range of physiological processes, including metabolism, cellular growth, and gene expression, its dysregulation can bring about devastating consequences to cell function, as implicated in a variety of diseases. As is the case with FD, the huge increase in cAMP level causes the cAMP-PKA-CREB pathway to be under perpetual stimulation via autocrine and paracrine actions of parathyroid hormone-related protein (PTHrP) by FD-derived BMSCs and osteoprogenitor cells [52,53]. High levels of PTHrP were detected both intracellularly and extracellularly, even in the late stages of osteogenic differentiation. Interestingly, studies have demonstrated a higher level of sensitivity and responsivity of GNAS-mutated cells of the osteoblastic lineage to hormonal stimulation than normal cells [54]. As serum levels of parathyroid hormone (PTH) are often elevated in FD patients, these mutation-bearing osteoblast lineage cells are under a sustained and prolonged exposure to PTH, and under such stress, have shown an overactivation of the Wnt/β-catenin signaling pathway [55,56,57].

The Wnt/β-catenin signaling pathway works in both synergistic and antagonistic manners to regulate osteogenesis in BMSCs; it may serve as a critical promoter of osteogenesis and a mediator of fundamental cellular activities, including growth, proliferation, differentiation, and apoptosis, or may hinder the osteogenic process by deterring osteoblastic maturation [58,59,60,61]. Consistent with these views, BMSCs derived from a FD patient or stimulated via PTHrP or WNT family member 3A (WNT3A) displayed an extreme prominence of early osteogenic markers, i.e., the bone matrix proteins COL1 and ALP and the transcription factors RUNX2 and OSX, but when overstimulated with high levels of PTHrP or WNT3A, a reduced maturation status was observed [62,63]. The role of the GNAS mutation in accelerating the early stages of osteogenic differentiation was further supported by robust expressions of early osteogenic markers in a tissue-specific and localized manner via the Cre recombinase-mediated activation of the GNAS mutation in transgenic mice in vivo and from BMSCs isolated in vitro [64]. This finding is of major mechanistic importance, as it confirms the stagnant state of mutant BMSCs in the early stages of the osteogenic lineage, as suggested by the prominent expression of markers characteristic of pre-osteoblastic cells.

These early osteoblastic markers carry potential to be translated into clinical applications. Elevated serum ALP levels in FD patients were reported to be positively correlated with FD recurrence and progression in multiple clinical studies, raising the possibility of ALP as a prognostic and diagnostic marker [65,66,67]. Additionally, the critical role of RUNX2 in suppressing osteoblast maturation to support the expansion of an immature osteoblast population in FD identifies RUNX2 as a probable candidate for FD diagnosis and a treatment target [68]. A recent study by Xiao et al. introduced the possibility of epigenetic involvement by revealing the direct participation of histone deacetylase in mediating the deacetylation of transcription factors, such as RUNX2 and tumor protein 53 (TP53), for suppressed osteogenesis and augmented proliferation of BMSCs in FD [69].

The intensified TGF-β/BMP signaling pathway is another mechanism perturbed that may contribute to the impaired osteoblast differentiation in FD. TGF-β has been reported to endorse the recruitment and proliferation of BMSCs and their differentiation into the early stages of the osteogenic lineage but conversely hinder its maturation into osteocytes and mineralization [70]. It is also known to exert a profound influence over the activation of fibrogenesis to stimulate ECM overproduction and is regarded the central mediator of the fibrotic phenotype involved in fibrotic diseases [71]. The overproduction of TGF-β in FD may account for the abundance of immature osteoblasts and excess extracellular matrix observed in FD lesions. TGF-β1, in particular, has been frequently detected in fibroblastic cells of FD [68]. It may be interesting to consider a possible association between the phenotypic characteristics and severity of the condition with TGF-β. Furthermore, other growth factors of the TGF-β family, including BMP2 and BMP4, have also been found dispersed throughout the osteoid and mineralized structures with some immunolocalization in fibroblast-like spindle cells in FD [72]. BMP2 has been implicated in the stimulation of ALP induction in pre-osteoblastic cells to promote bone formation, which suggests its contributory role in the elevation of ALP observed in both serum and FD samples [73].

#### 5.1.2. Impaired Osteoblast Maturation

Osteopontin, OPN, and osteonectin, ON are non-collagenous matrix proteins that appear in the mid and late stages of osteoblast differentiation. Normally, the deposition of OPN can be found ahead of the mineralization front of osteoid and at high levels in compact cortical bone, whereas ON exhibits an even dispersal throughout the osteoid and strong immunolocalization in the trabecular bone [29,74]. Immunostaining of FD lesions revealed comparative patterns of expression; OPN expression was mostly localized to the bone matrix, whereas ON expression was exclusively detected in fibroblast-like cells [47]. In accordance with the mineralization deficit, FD specimens demonstrated reduced levels of OPN and increased levels of ON that correspond with the immature and undermineralized nature of FD [29,75,76].

Both OPN and ON are reported to hold multifunctional roles in promoting mineralization and in the recruitment of osteoclasts and immune cells for bone remodeling, but based on histological interpretations, it can be presumed that OPN and ON have temporally and spatially varying expressions that are dependent on the stage of crystal precipitation for mineralization [77,78]. Although its exact function in bone formation is still unclear, the presence of OPN at the de novo mineralization front suggests OPN to be involved in the earliest stages of mineralization associated with the promotion of apatite crystal growth and the adhesion of cells to the extracellular matrix [79,80]. Given the substantial degree of de novo bone formation seen in FD lesions, it is expected for OPN to be strongly expressed, but rather its suppressed expression may explain the hindered mineralization in FD as a result of a comprised cell–extracellular matrix interaction and impaired crystal growth [81,82]. ON, on the other hand, is a relatively late-stage marker compared to OPN, which was found to be expressed by all cells residing in the stroma, including fibroblasts within the FD lesion. Past studies have reported ON to be a nucleator of hydroxyapatite crystal formation that is consistent with the lack of bone mineral hydroxyapatite at FD-affected sites [74].

Further studies have revealed a raised secretion of fibroblast growth factor-23 (FGF23) by mutated cells of the osteoblast lineage, particularly fibroblastic cells, osteoblasts, and osteocytes [83]. Increased FGF23 directly acts on renal proximal tubule cells: the sodium phosphate cotransporters for reduced renal phosphate reabsorption and the renal hydroxyvitamin D3 1α-hydroxylase for decreased active vitamin D synthesis [84,85]. In response to the lack of blood phosphorous levels and vitamin D levels, osteoblastic gene expression levels are indirectly suppressed, with osteoprotegerin (OPG) production levels increased for the inhibition of osteoblasts transiting into osteocytes [86]. Together, these findings support the claim that osteoblast maturation is impaired in FD, resulting in a reduced number of viable osteocytes and a decreased level of osteocytic activity necessary to achieve normal mineralization.

#### 5.1.3. Insufficient Mineralization

Insufficient mineralization is a key pathogenic factor for the skeletal aberrancies attributed to FD. The key proteins involved in the terminal stages of osteogenic differentiation and mineralization are namely BSP, OC, and sclerostin (SOST). Significantly weakened, or even absent, expression levels of these late osteogenic markers were observed in FD [75,87,88]. BSP and OC are mainly secreted via mature osteoblasts, whereas sclerostin is primarily produced by osteocytes to modulate bone remodeling activities [89]. The significant downregulation of these markers has been presumed to be induced in response to the substantial activation of the canonical WNT/β-catenin pathway [39]. A study led by Zhao et al. verified the relation between the late osteogenic marker OC and GNAS mutation expression using a FD transgenic mouse model, with which they successfully demonstrated the creation of fibroblast-like pre-osteoblasts and the development of a dysregulated and poorly mineralized woven bone matrix, both distinct histopathological manifestations of FD [64]. More importantly, the expression of these late osteogenic markers was shown to vary according to the severity of the condition at presentation, which suggests their potential use as a tool for FD diagnosis and prognosis.

In line with the abnormal osteoblastic maturation of the bone-forming mesenchyme, a delay in the rate of skeletal growth and maturation can be seen in FD. When compared to the normal ossification process, the formation of rudimentary bone resembling intramembranous ossification, with a flattened appearance of the bone-lining cells, was observed in the lesions [90]. In summary, cAMP overproduction via mutated BMSCs activates osteoinductive signaling pathways that stimulate and promote osteoblast proliferation and differentiation but deter maturation, resulting in dynamic changes in bone metabolic markers to yield altered mineral-to-matrix ratios. Despite the need for clarification on the many aspects concerning these pathways and mechanisms, the evidence points to the BMSCs as the initial and chief culprit intruding on the normal biology of healthy tissues for the development of FD.

### 5.2. Fibrous Matrix Deposition

#### 5.2.1. Delineating Fibroblastic Cell Population

Fibrosis is a major driver associated with the impedance of bone formation in FD. Fibrous dysplastic lesional cells have often been described as “fibroblast-like” by many researchers due to its shared morphological features and collagenized extracellular matrix-producing ability in spite of harboring molecular and cell surface markers of the osteogenic lineage [29,91,92]. A bone marrow fibrotic mouse model designed to mimic the fibrosis event that occurs in FD revealed distinct subpopulations of fibrocytes, among which a selected group negative for CD45 were presumed to be differentiating into myofibroblasts [92]. To this day, there is an enigma surrounding the cell type of these dysplastic mutant cells. However, upon findings with prior investigations, it may be inferred that a number of entities may be in a state of dynamic flux between stromal cells, fibrocytes, and myofibroblasts, amid osteoblastic cells [90,93,94,95]. These stellate-shaped stromal cells, fibrocytes, and myofibroblasts are believed to be critical contributors to the extracellular matrix deposition, as evidenced in many fibrotic events involving the bone marrow, lungs, liver, and kidneys. In such scenarios, these cells have been reported to express upregulated processing levels of the WNT/β-catenin- and TGF-β/BMP-mediated pro-fibrotic signaling pathways, which are key pathogenic mechanisms for fibrosis [96,97,98].

#### 5.2.2. Role of TGF-β

A recent study confirmed upregulated levels of the pro-fibrotic TGF-β via molecular analysis of GNAS mutant cells [99]. As previously mentioned, TGF-β is a key regulator for fibrogenesis that holds immense importance in promoting myofibroblast transdifferentiation, ECM deposition, and fibroblast growth and proliferation [100]. Alpha smooth muscle actin (α-SMA)-expressing myofibroblasts have been implicated in many fibrotic pathologic conditions due to their ability to not only promote contraction but also synthesize ECM proteins, like collagen and fibronectin [101]. An excessive contraction of myofibers and a persistent accumulation of ECM components are events that lead to fibrosis. This emerging evidence suggests that the vast number of myofibroblasts found in FD lesions may have derived from not just one but several origins, such as resident fibroblasts, osteoprogenitor cells, and pericytes, and even endothelial lineage cells [102,103]. Moreover, immune cells, such as macrophages and lymphocytes, are also believed to contribute to fibrogenic activation via the secretion of growth factors, chemokines, and cytokines, of which TGF-β is considered the key mediator of the fibrosis system [71]. In regard to the substantial amount of fibrosis present in FD and the upregulated expression of TGF-β by fibrous dysplastic cells, TGF-β may serve as an eligible candidate marker for detection, diagnosis, and treatment against fibrous tissue buildup.

#### 5.2.3. Sharpey’s Fiber Formation

In line with the induction of the pro-fibrotic signaling pathways, a robust increase in extracellular matrix molecules, i.e., COL1A1 (collagen, type I, alpha 1), COL4A1, and COL11A2, collagen crosslinking enzymes, i.e., PLOD2 (procollagen-lysine, 2-oxoglutarate 5-dioxygenase 2) and LOX (lysyl oxidase), and adhesion molecules, including LIMS2 (LIM zinc finger domain containing 2), ITGA6 (integrin subunit alpha 6), and PXN (paxillin) was observed in FD-derived cells [99]. Regarded as a distinct pathological trait of FD is the presence of high-density Sharpey’s fibers situated between the stroma and the trabeculae. The nature of these fiber bundles is rather unique in that it is a matrix of the connective tissue that is highly collagenized, non-mineralized, and mainly composed of collagen type I [55]. Histologic analysis demonstrated a focal localization of the extracellular matrix protein, periostin (POSTN), to Sharpey’s fibers of FD and implicated its interaction with type I collagen to form and maintain these structures [104]. Enhanced levels of cytokines found in FD as a result of aberrant cAMP production provides the conditions conducive for POSTN expression, often triggered by cytokines like TGF-β, IL4, and IL13 [105]. Other affected genes include FGF8 [55] and ADAMTS2 (ADAM metallopeptidase with thrombospondin motifs 2) [106], of which their overexpression in combination with MMP3 may contribute to the degradation of the extracellular matrix that, in turn, paves the way for Sharpey’s fibers to occupy void spaces in FD.

### 5.3. Bone Remodeling Imbalance

Considered an essential component in the pathology and biology of FD is a process called bone remodeling. It is crucial for maintaining skeletal integrity, regenerating and repairing microdamages, and monitoring mineral homeostasis [27]. This process occurs asynchronously upon or within any bone surfaces at multiple focal sites in the mature skeleton. The trabecular bone typically exhibits a higher rate of metabolic activity than the cortical bone [107]. Bone remodeling entails the coherent, orchestrated action across two major cell types: the osteoblasts and the osteoclasts. The detection of microdamages of the bone transmits signals for the recruitment of monocytes or macrophage osteoclast precursors to the specific site from the circulation via the capillaries, bone marrow, or the bone’s surface. These precursor cells undergo differentiation into multinucleated bone-resorbing osteoclasts by macrophage colony-stimulating factor (M-CSF) and receptor activator for nuclear factor κβ ligand (RANKL) stimulation and cause them to gain important functions in bone remodeling regulation and skeletal health maintenance [108].

#### 5.3.1. IL6 and Osteoclastogenesis

IL6 is believed to hold pathogenic importance in fostering enhanced osteoclastogenesis and bone resorption responsible for the phenotypic expression of this disease. Accrued studies performed on active FD patients demonstrated increased levels of cytokine IL6 in the blood serum that showed strong positive correlations with the clinical disease activity, the frequency of relapses, and the severity of disease progression [109]. Scrutiny of these lesions indicated a direct association between excess cAMP levels and cFOS expression by FD-derived cells and IL6 overproduction; the increased level of IL6 was shown to promote osteoclast recruitment and bone resorption at the periphery and in the interior areas devoid of bone [110]. Additionally, IL6 is believed to have the effect of multiplying the number of osteoclast nuclei per cell by nearly two to three times [111]. Corroborating this notion, a high level of IL6 secretion was detected in the BMSCs, stellate stromal cells, and osteoblasts, and even in the bone-lining cells of the fibrous dysplastic tissue, adjacent TRAP-positive osteoclasts pervading both the external and internal areas of the lesion [28,112]. Furthermore, immune cells, like macrophages and dendritic cells (DCs), were also found to significantly express RANKL, which was presumed to contribute towards the secretion of proinflammatory cytokines for the elevated activity of osteoclasts [18,32,76,113]. An amplified effect of this phenomenon may be obtained via autocrine and paracrine mechanisms mediated via the self-perpetuated extracellular release of vesicular RANK [114,115]. It is intriguing to note that in consort with increased serum OPG levels, a huge upsurge in serum RANKL of up to 16-fold higher than OPG was observed in FD patients [116]. Thus, despite an overexpression of OPG, the magnitude of change in the levels of RANKL yields a greater RANKL/OPG ratio that negates the actions of OPG for osteoblastic differentiation and osteoclastogenesis inhibition.

#### 5.3.2. Other Cytokines and Chemokines

More recently, molecular profiling of FD lesions suggested the contribution of numerous cytokines and chemokines in promoting osteoclastogenesis. Cytokine genes, such as interleukins (ILs), i.e., IL1A, IL1B, IL6, IL7, IL11, IL12B, IL13, IL15, IL23, and IL24, interferons (IFNs), i.e., IFNA2, IFNA7, and IFNA17, thymic stromal lymphopoietin (TSLP), colony-stimulating factor 2 (CSF2), and erythropoietin (EPO) were significantly upregulated [55,83], in addition to chemokine genes, such as CC motif chemokine ligand 2 (CCL2) and CXC ligands (CXCLs), i.e., CXCL1, CXCL13, and CXCL2 [30]. Under such conditions, an abundance of osteoclasts and the aggressive level of osteoclastic activity in FD implicates a greater likelihood for a proinflammatory M1-type macrophage presence, which produces several types of cytokines, such as TNF-α, IL1-β, and IL6, to activate osteoclasts and promote bone resorption [117]. Factors reported to be involved in the stimulation of bone turnover include progesterone, PTHrP, IL1, IL11, IL17, and TNF-α, many of which are overexpressed in FD [118]. Hyperactive bone turnover has been associated with several disorders, such as osteoporosis, renal osteodystrophy, Page’s disease, and osteopetrosis [119]. With respect to the roles of cytokines and chemokines in modulating immunologic homeostasis and inflammatory events, their overexpression has been expected to make a substantial contribution to the accelerated turnover rate and the disrupted balance between osteogenesis and osteoclastogenesis that, in concert, causes the destructive patterns of bone erosion seen in FD.

### 5.4. Bone Pain

Bone pain is considered one of the more problematic immediate symptoms experienced by FD patients, and for this reason, it has been a matter of interest in the past decade for search of better treatment options. Clinical studies have revealed discrepancies in the amount of pain felt, from minimal to excruciating, by patients of different age groups and FD subtypes [120]. However, a general consensus has been reached in that FD-related bone pain tends to exacerbate with age [121]; approximately 81% of adult FD patients reported bone pain compared to 49% amongst FD children [122]. One study indicated monostotic FD to be more susceptible to severe pain [123], whereas a more recent study suggested differently, with findings that demonstrate higher pain scores amongst polyostotic than monostotic FD patients [124]. More consistent results were obtained with regard to pain sites, with the most pain originating from the lower extremities and the ribs than the upper extremities and craniofacial areas [122,123,125,126]. Owing to the unclear relationship between pain intensity and FD burden, its precise pathophysiology remains elusive. Plausible postulates have been proposed based on other bone diseases and cancer. Some have deduced intensive osteolysis and bone remodeling to be causative factors of bone pain [127,128]. However, based on clinical findings, no direct correlation between osteolysis alone and bone pain was discovered, leading others to suspect the involvement of neural factors.

#### 5.4.1. Nociceptive Factors

Bone pain has long been studied in bone cancer patients, and recent studies have indicated two components, both a nociceptive and neuropathic factor, for inducing pain [121]. Pro-nociceptive compounds include products of inflammation, such as protons, prostaglandin, cytokines, chemokines, and reactive oxygen species (ROS) [129,130]. In an osteolytic setting, the activation of osteoclasts via RANKL expressed by stromal cells establishes an acidic environment through proton extrusion that excites nociceptors, including the ASICs and TRPV1 channels of nerve fibers, which trigger bone pain [131,132]. Cancer cells and stromal cells are believed to contribute to the influx of nociceptive compounds via the release of IL6, GM-CSF, endothelin, TNF-α, proteases, and nerve growth factor [133]. Such environmental conditions are comparable with FD surroundings manifested through intensified osteoclast bone erosion and increased production of proinflammatory mediators, i.e., cytokines, chemokines, ROS, and MMPs, by mutant osteogenic cells that makes it very likely for pain to be generated under similar manners.

#### 5.4.2. Neuropathic Factors

Another potential mechanism involves neuropathic pain generated via the ectopic sprouting of sensory and sympathetic nerve fibers and peripheral and central sensitization derived by plasticity changes in the central nervous system along with bone cancer progression [134,135]. Such an occurrence was also observed in cases involving osteoarthritis and intervertebral disc pain [136]. Tumors have been known to damage the most distal aspect of sensory nerve fibers leading to its degeneration, thereby providing reasons for the chronic and acute pains. Pain can also originate from nerve distortion and compression from mechanical strains caused by excessive bone remodeling-related infrastructure instability and reduction in inherent strength [137]. Although its precise role in FD is yet to be confirmed, analogies that have been made based on other bone-related diseases have identified various nociceptive and neuropathic factors as likely contributing factors to bone pain.

Together, these findings suggest the heterocellularity and complexity associated with FD’s pathogenesis. A FD-controlled environment contains populations of diverse cell components, i.e., mesenchymal stem cells, osteogenic cells, immune cells, osteoclastic cells, fibroblastic cells, and smooth muscle cells, that communicates both intra- and inter-cellularly to orchestrate the events seen in FD (Figure 2). Despite mutated BMSCs being the source of the perturbation, disrupted pathways and impaired functions in combination with crosstalks with different cell populations appear to give rise to the pathological conditions.

## 6. Current Treatment Prospects

The development of therapies has been hampered by two major obstacles: a lack of clinical and research knowledge and scarcity of expertise. However, with the incorporation of advanced analytical tools and sophisticated bioengineering techniques, emerging evidence with respect to the mechanism of action at the genetic, molecular, and cellular level has largely contributed towards the significant evolvement of our understanding on FD pathogenesis over the last decades. With knowledge on the precise nature of etiologic and pathophysiologic factors, increasing efforts are now being placed on developing prospective medical therapies capable of either eliminating the genetic source of the problem or altering the course of disease progression. Despite surgical management being the most effective treatment option to date, the invasiveness of the procedure carries risks for pain aggravation and surgical complications of excessive resection or unsuccessful reconstruction using bone grafts and implants. Hence, more non-surgical options are being investigated and tested for treating, particularly involving young FD patients. Current and potential treatments for FD targeting different aspects of the disease using small molecules, antibodies, and CRISPR-based gene editing technologies are summarized in Table 2.

### 6.1. Modulation of Osteoclastogenesis

The core pathological change in FD implicates perturbations of bone formation and resorption caused by aberrant osteoblastic and osteoclastic activities. An excessive release of cytokines and chemokines and the subsequently escalated bone remodeling process are the determinant factors that induce the pathological changes and clinical manifestation. In particular, osteoclast recruitment and its activation is closely related to patterns of bone resorption observed in FD. Interventions that regulate osteoclastic activities by rendering it inactive are effective measures that have been demonstrated to improve the prognosis of FD.

#### 6.1.1. Bisphosphonates

Treatments using small molecule-based drugs, such as bisphosphonates, in FD patients experiencing an aggressive progression of anomalous bone development and/or persistent pain are considered a first-line therapy. With options of oral or intravenous administration, potential drug candidates include alendronate, pamidronate, and zoledronate [138]. Their mechanisms of action involves the direct interference of osteoclastic activity by inhibiting the mevalonate pathway and preventing the isoprenylation of GTP-binding proteins [139]. It exhibits the effect of suppressing bone resorption and decreasing inflammatory responses from the osteolytic process. In doing so, it is believed that bisphosphonates may be beneficial in the regeneration of bone for increasing the bone mineral density (BMD) at lesion sites [140]. It has also demonstrated central and peripheral antinociceptive effects by suppressing the production of pro-inflammatory cytokines, neuropeptide Y, and prostaglandins.

Several open and randomized clinical studies have been performed using bisphosphonates in FD/MAS patients that demonstrated a universal improvement of bone resorption, with some radiographic improvements that indicated regenerated lesion sites and/or cortical thickening [141]. Its effect on bone pain is, however, questionable. Orally administered alendronate at high doses were revealed to be less effective against bone pain than intravenously administered pamidronate or zoledronate [142,143,144]. Considering the minimal complications and potential beneficial effects on bone resorption and pain, the use of intravenous bisphosphonates, such as pamidronate or zoledronate, is considered best practice in the management of FD/MAS to date, according to the International FD/MAS Consortium [145].

#### 6.1.2. RANKL Inhibitors

Scarce studies on other anti-resorptive agents have made them less reliable for use. In more recent years, RANKL inhibitors have been gaining recognition as potential therapeutic agents to treat FD-associated skeletal progression and bone pain. Increased levels of RANKL are frequently observed in stromal cells, including cells of the osteogenic lineage, in the microenvironment of FD patients, and the disrupted balance of the RANKL-to-OPG ratio exacerbates this disease through enhanced recruitment and stimulation of osteoclastic activities, making RANKL inhibition a potential target for therapy.

Superior results over bisphosphonates were obtained using a human monoclonal anti-RANKL antibody, denosumab, that demonstrated a cessation of FD progression [133,146,147]. Furthermore, animal studies and off-label clinical use have demonstrated encouraging outcomes of anti-osteoclastic effects, reduced FD activity, and bone growth after RANKL neutralization [127,147]. To date, two clinical trials registered in Clinicaltrials.gov (accessed on 12 October 2023) are being performed: one to assess the efficacy of denosumab against FD lesion progression in children, and the other to test the efficacy of denosumab in improving the clinical, radiological, and biochemical manifestations of FD lesions. Severe hypercalcemia development following discontinuation is a major complication associated with its use that requires special attention to establish optimal doses and duration of therapy.

A recent study developed a small molecule-based RANKL inhibitor, AS2676239, that was able to emulate the biological effects of denosumab in suppressing osteoclastogenesis to halt or attenuate the disease progression and allow for bone remineralization at lesion sites [148]. Although varied levels of bone turnover increase in response to treatment discontinuation was observed, the high success rates of the RANKL antibodies and AS2676239 makes it an attractive option for the prevention and management of this disease, especially in individuals with early-onset FD [127].

#### 6.1.3. IL6 Inhibitors

IL6 has been extensively studied in relation to FD and its pathologic importance in inducing osteoclastogenesis and impairing osteoblast differentiation. Considering that increased levels of IL6 production were detected in various types of cells derived from the lesion, it serves as an appealing target to modulate bone resorption and bone pain in FD patients. Tocilizumab, an antagonist of IL6, has been applied to systemic autoimmune and chronic inflammatory disorders, such as rheumatoid arthritis and systemic sclerosis, due to its therapeutic effect of counteracting cytokine storms [149,150]. Despite previous studies identifying tocilizumab as a pharmacological intervention to target pain and bone turnover in FD/MAS patients, a recent clinical study has revealed that the effects of this drug is rather insignificant [151]. At present, IL6 antagonists are not suggested in the treatment of patients with FD, but due to mosaic levels of IL6 being found in FD patients, the efficacy of IL6 antagonists deserves further exploration directed towards a larger subset of individuals naïve to bisphosphonate treatments.

#### 6.1.4. CSF1R Inhibitors

Monocytes and macrophages are recurring entities that are important to FD biology that makes it amenable to therapeutic targeting. Macrophage colony-stimulating factor (CSF1) is an important growth factor necessary for the activation, differentiation, and function of monocyte-derived macrophages [152]. High levels of CSF1 expression can be identified in all monocyte-derived macrophages and bone-marrow derived macrophages, and its deliberate overexpression demonstrated increased monocytosis and proliferation of macrophage populations [153,154]. Administration of an anti-CSF1R antibody led to the rapid ablation of interstitial macrophages and osteoclasts, which suggests its potential use to manage hyperactivated macrophages/osteoclasts and overstimulated resorption activities [155,156]. With the prospect of surveilling cells of the monocyte–macrophage lineage, there is an emerging interest in the use of CSF1R blockers or antagonists to target lesion growth in FD patients.

### 6.2. Blockade of Neural Sensitization

Bone pain is regarded as one of the major symptoms for which FD patients seek medical attention. Since the pain appears to be directly related to the pathological sprouting of sensory and sympathetic nerve fibers that lead to peripheral and central sensitization and persistent pain symptoms, currently available therapeutic interventions typically involve inhibiting the nerve growth factor (NGF)/tropomyosin-related kinase A (TrKA) interaction and suppressing the activation of transient receptor potential cation channel subfamily V member 1 (TRPV1) [157]. Despite the controversy regarding its clinical safety and efficacy, these treatments have demonstrated remarkable improvements in musculoskeletal pain, making it the most tangible target for designing therapeutics for FD-associated pain.

#### 6.2.1. NGF/TrKA Inhibitors

NGF, a neurotrophin secreted by immune, inflammatory, and even non-immune cells, like fibroblasts and T-cells, often upregulated in response to injury and inflammation, plays a pivotal role in activating and sensitizing nociceptors for the feeling of pain [158,159,160]. Treatment of anti-NGF or TrkA inhibitors showed beneficial effects against ectopic TrkA+ nerve sprouting and peripheral sensitization to manage chronic pain associated with cancer, autoimmune arthritis, and inflammatory disorders [161,162,163,164]. Although no distinct correlation has been made that links NGF to the bone pain experienced by FD patients, based on the pathological mechanism observed, inhibiting NGF or TrkA may be effective in attenuating the ectopic reorganization of nerve fibers within and surrounding the lesion and mitigating the sensitization of these nerve fibers.

#### 6.2.2. TRPV1 Antagonists

Similar in concept with the NGF and TrkA inhibitors, TRPV1 antagonists are being deemed as a viable therapeutic agent to treat pain and sensitization associated with injury and inflammation. Modern drugs are being developed that target TRPV1 to treat a variety of pathological conditions ranging from cancer, osteoarthritis pain, atopic dermatitis, and even diabetic neuropathic pain [148,165,166]. TRPV1 is considered a critical calcium-permeable ion channel expressed on immune cells that are activated by vanilloids, like capsaicin, to depolarize nociceptive neurons for the perception of pain [167,168]. It may potentially be activated in FD by the sudden drop in pH from enhanced osteoclastic activities that, in turn, activate TRPV1. Amongst the TRPV channels, only TRPV1 has been reported to be stimulated from pH changes [169]. Under the acidic conditions of FD, acid-sensing ion channels (ASICs) expressed on the surface of nociceptors may also be an attractive target. ASIC1 and ASIC3 have been demonstrated to contribute to bone pain experienced by rheumatoid arthritis patients, which was effectively relieved with its blockade [170,171].

### 6.3. Inhibition of Gsα

A high percentage of patients, albeit not all, yielded consistent data for the presence of a missense mutation of the alpha subunit of the G protein, resulting in an overproduction of cAMP due to hyperstimulated adenylyl cyclase. Approaching the fundamental foundations of this disease means either inhibiting or silencing the problematic G protein using small molecule inhibitors or gene-editing technologies. With advances in the molecular understanding of G protein activation, novel peptide and non-peptide-based drugs are being actively developed to target diseases involving G proteins and G protein-coupled receptors. Moreover, these diseases tend to be hereditary, making the use of CRISPR gene editing and silencing technology an attractive option to directly alter the defective genetic component for a complete elimination of the source of the problem. It is expected to prevent and provide relief to most, if not all, clinical symptoms of FD, including disease development, progression, and pain in patients.

#### 6.3.1. G Protein Antagonists

Several steps exist in the activation of the G protein, thus providing opportunities to target the G protein α and βγ subunit interface and receptor –G protein coupling [172]. Available options include modified guanine nucleotides that irreversibly bind to the G protein’s α subunit to inactivate the GTP into GDP and receptor- or G protein-derived peptides and nonpeptides that interfere with the interaction between the receptor and G protein, amongst which suramin is considered the lead compound for manipulating the G protein-regulated signaling mechanism [173]. It demonstrates inhibitory activities via three known mechanisms: (1) preventing pre-bound GDP dissociation, (2) blocking the G proteins’ α and βγ subunit interface, and (3) inhibiting the formation of ternary agonist-receptor–G protein complexes [170]. Treatment of suramin sodium in vitro presented with promising results, with reduced proliferative activity and decreased cAMP levels via FD patient-derived BMSCs. More potent drug candidates may include suramin analogues, such as NF449 and NF503, that encompass subtype selectivity for Gsα [174,175].

#### 6.3.2. GNAS Gene Editors/Silencers

Gene-editing tools, like CRISPR/Cas9, are being used to mediate the silencing of the GNAS gene in various tumor models, including pancreatic, appendiceal, and liver cancer [176,177]. Given the rarity of FD, there is still a lack of robust animal and human models that can be used to comprehend the mechanisms of FD. Novel strategies, using gene-editing technologies, have been utilized to engineer various types of genomic alterations from single gene knockouts to single-base nucleotide modifications in in vitro cell cultures and in vivo animal models to harbor either a R201H or R201C missense mutation of the GNAS gene [11,178]. In a similar manner, targeted silencing or knockdown of the GNAS gene within these FD models resulted in a significant recovery and restoration of bone formation functions [91,179]. It is expected for more advanced gene-editing technologies, i.e., CRISPR base editing and prime editing, that enable precise base repairs from the single-nucleotide level to pave the way for precision targeted gene therapy [180].

**Table 2 ijms-24-15591-t002:** Current and potential therapies indicated for the treatment of FD.

Agent	Target	Action	Type of Study	Efficacy			Potential Complications	Refs.
				Bone Turnover Decrease	Bone Pain Reduction	Radiographic Improvement		
**Current therapies**								
Bisphosphonates 1. Alendronate	Bone resorption/bone formation	Osteoclast function and proliferation inhibition, osteoclast apoptosis, osteoblast differentiation, and activity promotion	CR (*n* = 1)	+	+	+	No adverse effects	[181]
			CR (*n* = 1)		+	+	No adverse effects	[182]
			Phase II (*n* = 40)	+	-	-	Nausea and vomit (pediatric)	[142]
2. Pamidronate			CS (*n* = 9)	+	+	+	Transient mineralization defect	[183]
			CS (*n* = 9)	+	+		No adverse effects	[184]
			CS (*n* = 13)	+	+	+	No adverse effects	[185]
			Phase II (*n* = 58)	+	+	+	Transient fever	[143]
3. Zoledronic acid			CS (*n* = 11)	+	+		No adverse effects	[186]
			CS (*n* = 10) peds.	+	+	+	Hypocalcemia	[144]
			CS (*n* = 7)	+	+	-	Transient fever and myalgia	[187]
Anti-RANKL (denosumab)	Bone resorption	Osteoclast activation suppression	CR (*n* = 1)	+	+	+	Hypophosphatemia and secondary hyperparathyroidism	[127]
			CS (*n* = 12)	+	+		No adverse events	[188]
			CR (*n* = 2)	+	+	+	No adverse events	[188]
			Phase II (*n* = 8)	+	+	+	Hypercalcemia post-discontinuation	[189]
			Phase II (*n* = 15 estimated)	NA	NA	NA	NA	*
			Phase IV (*n* = 82 estimated)	NA	NA	NA	NA	*
Anti-IL6 (tocilizumab)	Bone resorption	RANKL production suppression	CR (*n* = 1)		+		No adverse events	[190]
			Phase II (*n* = 16)	-	-	-	No adverse events	[151]
**Potential therapies**								
CSF1R inhibitors	Bone resorption	Macrophage formation inhibition						[191]
Anti-NGF/TrKA (tanezumab)	Bone pain	Nerve sprouting and sensitization blockade						[192]
TRPV1 antagonists		pH-sensitive neurons’ blockade						[193]
Gsα inhibitor (suramin sodium)	G protein	Mutant Gsα inhibition						[194]
Gsα gene editor		Gsα mutation correction						[195]

Abbreviations: CR, case report; CS, case series; NA, not applicable; RANKL, receptor activator of nuclear factor kappa β ligand; IL6, interleukin 6; CSF1R, colony-stimulating factor 1 receptor; NGF, nerve growth factor; TrKA, tropomyosin receptor kinase A; TRPV1, transient receptor potential cation channel subfamily V member 1. * Ongoing clinical study registered in Clinicaltrials.gov (accessed on 12 October 2023).

## 7. Concluding Remarks

Losses in tissue and cellular identity and the resultant disordered interplay across various cell types has been identified as the fundamental cause of the pathological process underlying FD. Understanding the tissue and cell biology of FD, thus, may be of essence in uncovering the pathological mechanisms involved. Yet, studies of cellular interactions in bone-related diseases have posed a major challenge. However, modern tools coupled with in vitro experimental approaches have provided valuable insights into the heterocellular complexity of FD.

This review discussed the implication of mutated BMSCs and their interactions across various cell types in pathologic conditions, including impaired bone formation, aberrant bone metabolism, and bone pain. It is worthy to notice the widespread impact of a single point mutation in the GNAS gene on various physiological processes, such as extracellular matrix synthesis, mineralization, and degradation. The extensive involvement of mutated BMSCs in the clinical manifestations of FD presents a promising therapeutic opportunity. By targeting BMSCs or in tandem with the associated cells mentioned, it may be possible to delay or even inhibit lesion growth.

Fortunately, advanced technologies are being developed and new approaches applied to address the challenges inherent in the diagnosis and treatment of such rare genetic diseases. Genomic innovations have allowed for precise screening and diagnostic modalities to identify novel causative variants associated with this disease, and the unprecedented genomic manipulability together with cell engineering efforts has brought rare disease modeling and therapeutic discoveries closer to fruition than ever before.

Of note, FD has been in the spotlight in recent years due to a critical demand for therapeutic options capable of altering the course of this disease. Fortunately, there has been significant progress in furthering our understanding of the underlying pathologic basis and biological processes behind this disease to unveil the crucial target genes for prospective therapeutic development. More is yet to come, with the establishment of improved disease models capable of mirroring the pathologic and clinical manifestations of FD, which is expected to open new opportunities for exploring its niche environment and mechanisms to screen the effects of different therapies.

## Figures and Tables

**Figure 1 ijms-24-15591-f001:**
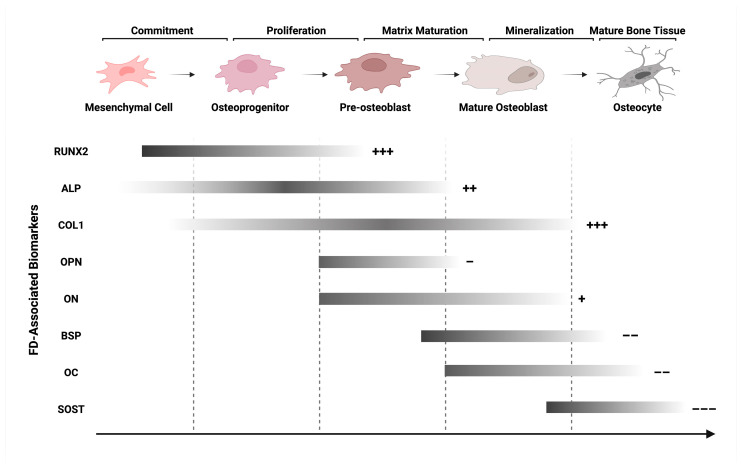
The biomarker expression profile of the osteoblast lineage in FD. Genes representing osteoblast differentiation, maturation, and mineralization, along with their putative changes in expression. Early committed BMSCs in FD express high levels of RUNX2 for increased proliferation, ECM overproduction, and osteoblast maturation inhibition, followed by ALP and COL1 expression by pre-osteoblasts for hyperactive alkaline phosphatase activity and active collagen deposition. Transitioning into the maturation phase, a decreased expression of OPN, a modulator of apatite crystal formation, and an increased expression of ON, a crystal nucleator and recruiter of immune cells, contribute to the immature fibrous matrix’s development. Towards mineralization, a dramatic decrease in BSP, OC, and SOST, late stage markers of osteoblast differentiation, lead to altered mineral metabolism and suppressed bone formation. The graded bars correspond to the relative expression levels of genes at the indicated stage of differentiation, and the +/− signs to the right of the bars denote the direction of change, i.e., up- or down-regulation, and the perceived magnitude of change in expression levels by the number of signs used. Abbreviations: RUNX2, runt-related transcription factor 2; ALP, alkaline phosphatase; COL1, collagen type 1; OPN, osteopontin; ON, osteonectin; BSP, bone sialoprotein; OC, osteocalcin; SOST, sclerostin. Created with BioRender. All rights reserved. Modified from the studies published by the authors of [47,48].

**Figure 2 ijms-24-15591-f002:**
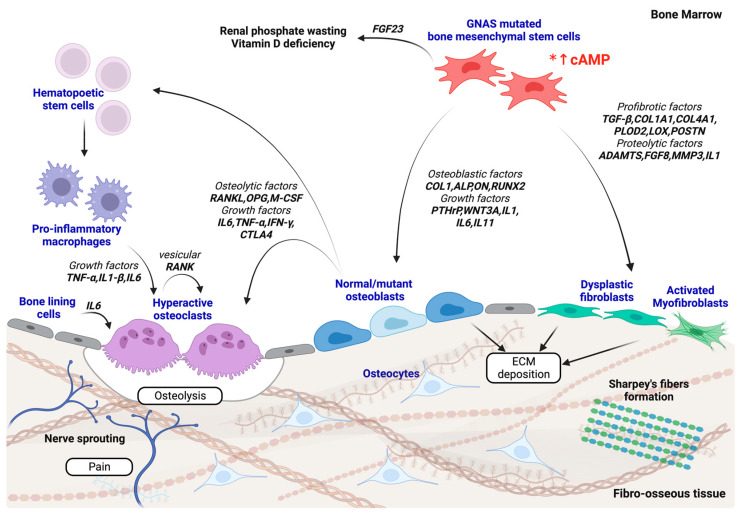
A schematic representation of the hypothetical FD pathology mechanism. In the FD bone niche, GNAS-mutated BMSCs lead to the constitutive overproduction of cAMP (marked with an *) that, in turn, affect a number of downstream target genes responsible for mediating various cellular activities involving the proliferation, differentiation, and maturation of osteoprogenitor cells, bone metabolic events, such as osteoclastogenesis, and extracellular matrix accumulation. GNAS-mutated BMSCs release osteoblastic factors (such as COL1, ALP, ON, and RUNX2) and growth factors (including PTHrP, IL1, IL6, IL11, and WNT3A) that lead to aberrant proliferative activity and early osteogenic differentiation into immature osteoblasts toward the FD phenotype. These immature osteoblasts release osteoclastogenic factors (including RANKL, OPG, M-CSF, IL6, TNF-α, IFN-α, and CTLA4) that induce proinflammatory macrophage and osteoclast differentiation. An upsurge of inflammatory cytokines via autocrine and paracrine mechanisms contribute to the hyperactivity of osteoclasts, resulting in pathological osteolysis and microarchitectural deterioration. Overexpression of FGF23 by the BMSCs causes renal phosphate wasting and vitamin D deficiency for debilitated deficits of mineralization. The production of profibrotic factors (including TGF-β, COL1A1, COL4A1, PLOD2, LOX, and POSTN) and proteolytic factors (such as ADAMTS, FGF8, MMP3, and IL1) stimulate dysplastic fibroblasts, myofibroblasts, and osteoblasts for the excessive deposition of the ECM for fibrous tissue formation, as well as Sharpey’s fiber development. Nerve sprouting and stimulation of nerve fibers by nociceptive and neuropathic factors generate bone pain. Together, dysfunctional bone formation and metabolism along with fibrotic events all contribute to FD development. Abbreviations: ALP, alkaline phosphatase; ON, osteonectin; RUNX2, runt-related transcription factor 2; PTHrP, parathyroid hormone receptor protein; WNT3A, wingless-type MMTV integration site family member 3A; RANKL, receptor activator for nuclear factor κβ ligand; IL1-β, interferon 1-beta; IL1, interleukin 1; IL6, interleukin 6; IL11, interleukin 11; TNF-α, tumor necrosis factor alpha; IFN-γ, interferon gamma; CTLA4, cytotoxic T lymphocyte-associated protein 4; M-CSF, macrophage colony-stimulating factor; COL1, collagen type 1; COL1A1, collagen type 1 alpha 1 chain; COL4A1, collagen type 4 alpha 1 chain; LOX, lysyl oxidase; PLOD, procollagen lysine 2-oxoglutarate 5-dioxygenase; POSTN, periostin; FGF8, fibroblast growth factor 8; FGF23, fibroblast growth factor 23; ADAMTS, a disintegrin and metalloproteinase with thrombospondin motifs; MMP3, matrix metalloproteinase 3. Created with BioRender. All rights reserved.

**Table 1 ijms-24-15591-t001:** GNAS somatic variants associated with FD.

DNA Nucleotide Modification	Predicted Protein Modification	Reference Sequences
c.601C>T	p.Arg201Cys (R201C)	NM_000516.7 NP_000507.1
c.601C>G	p.Arg201Gly (R201G)	
c.601C>A	p.Arg201Ser (R201S)	
c.602G>A	p.Arg201His (R201H)	
c.602G>C	p.Arg201Pro (R201P)	
c.602G>T	p.Arg201Leu (R201L)	
c.679C>A	p.Gln227Lys (Q227K)	
c.680A>T	p.Gln227Leu (Q227L)	
c.680A>G	p.Gln227Arg (Q227R)	
c.681G>T	p.Gln227His (Q227H)	

Abbreviations: R, arginine; Q, glutamine; C, cysteine; H, histidine; G, glycine; S, serine; P, proline; L, leucine; K, lysine. Reproduced with permission from the authors of [25].

## Data Availability

Data sharing not applicable to this article as no datasets were generated or analyzed during the current study.

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
