# Peer review of "A Rare Skeletal Disorder, Fibrous Dysplasia: A Review of Its Pathogenesis and Therapeutic Prospects"

_ijms, 2023, doi:10.3390/ijms242115591_

Round 1
Reviewer 1 Report
The manuscript is a review article on the pathogenesis and most recent therapeutic approaches of fibrous dysplasia.
No similar comprehensive summary on the knowledge about the disease has been published in the recent past. The review is based on a huge number (177) of references sorted as they appear in the text. Of the references 17 were published in 1 year, 62 in 5 years and 86 are dated back to the last 10 years.
The manuscript follows the instructions of the journal, the abstract word count is under the limit, the front and back matter contains the information required.
Major issues:
The manuscript contains 2 figures, of which one is a modified version of similar figures from Ref 47 and 48 as cited in the caption. Is there a permission from the publishers and authors of the above publications?
Besides, there are a lot of information about the expressions of several proteins and factors during the osteoblast differentiation, it may be worth to include some more in Fig 1.
The title of the review indicates that it covers the pathogenesis and the up-to-date therapy of FD.
Of the 15 pages – excluding the back matter and references – 11 pages are about the pathogenesis of FD in particular detail and 3 pages are about the treatment that only touches the new therapeutic modalities. The concluding remarks summarize the review only in broad terms and predict that new diagnostic and therapeutic (genomic innovation mentioned here) prospects are to be expected. Summary of recent findings on pathogenesis were not addressed here.
It would be of interest to expand the sections on therapy towards new promising modalities and their clinical relevances, side effects or even preliminary results from clinical trials, if any.
In the section on treatments, there is a citation to Ref 142 about the recent guideline on practice management. It may also be of interest and would help in the practice to draw an algorithm on the standard treatment steps and potential second line therapies according to the clinical manifestation of FD.
Minor issues:
An abbreviation list may be advised to be included, and they should always be explained the first time they appear in the text.
In Section 4 (row 176): IL-6 is not expressed, but produced/secreted and Ref 32/33 cited here do not have information on this, only citations in their reference sections.
In Section 5.1.2.: Is greater presence of ON over OPN exclusive for only fibroblast-like cells in FD? It would be worth, if OPN may be included in Fig 1, as well.
Is decreased OPN the only factor responsible for the lack of mineralization in FD?
Figure 2 is a very helpful and comprehensive summary of the disease pathogenesis, it may be advised to highlight or encircle the cells involved in the process (osteoblast, osteoclast, BMSCs) to make it easier to differentiate between the cells and the factors/proteins secreted by them.
Row 238 (Fig 1 caption): „suggested by findings in the present review” - ?
Segmentation of (too) long sentences is advisable at several places in the text to make them easier to read and interpret (e.g. see rows 118-122, 179-182, 425-430, 469-474, 524-527). (One might keep the "not more than 40 words in one sentence" "rule” in mind.)
There are some mistypes and grammatical errors, listed below:
Typos:
row 54 (monostatic- monostotic)
row 85 (G-protein – Gs-protein)
row 129 (seldomly – seldom or rarely)
row 140 (rule in the possibility for - … of)
Grammar:
row 146-149 (explain for the patients ...)
row 198 (entirely appreciate …)
row 215 (considered - considering)
row 228 (facets – steps?)
row 229 (any minute – any)
row 233 (manifest into – leads to)
row 258 (even into – even)
row 273 (stages osteogenic – stages of osteogenic)
row 296 (surfeit – not suitable in this context - overproduction )
row 300 (characteristic –characteristics, with measurements of TGFb – with TGFb)
row 329 (expression – production/secretion)
row 374 (expressing negative for CD45 – negative for CD45)
row 406 (considered – considering)
row 415 (which their overexpression, of which overexpression)
row 419 (considered – considering)
row 433 (conducive – responsible)
row 483 (debatable ?)
row 484 (otherwise – differently)
row 487 (the ribs more so than .. – the ribs than ..)
row 505 (heightened – increased)
row 618 (panoply - not suitable in this context – various types of …)
Reviewer 2 Report
Dear Authors,
I enjoyed vera lot this paper reading as it covered all the aspects of fibrous dysplasia and I especially liked the genetics approach with very clear definitions. It's worth being published as it is. Great hearing about all the genomic innovation.
Best regards.
Author Response
I am delighted to hear that you enjoyed reading our manuscript. Thank you for supporting the publication of this manuscript.Reviewer 3 Report
Fibrous dysplasia is a disorder where normal bone and marrow is replaced with fibrous tissue, resulting in formation of bone that is weak and prone to expansion. As a result, most complications result from fracture, deformity, functional impairment, and pain. Treatment in fibrous dysplasia is mainly palliative and is focused on managing fractures and preventing deformity. There are no medications capable of altering the disease course. This is an excellent review on the topic of fibrous dysplasia as it summarizes the pathogenesis as well as the emerging treatment options for this devastating rare bone disease. Specially, I like both figure 1 & 2. Figure 1 depicts the FD-associated bone biomarker and it’s i clinically relevant. For figure 2, it nicely summarizes the hypothetical FD pathogenesis. Just a small suggestion, is it possible to tabulate these emerging treatment prospects by listing out in what developmental stage for each option (e.g., pre-clinical, various phase of clinical trials).
excellent writing
Author Response
Thank you for your positive comments! We are very glad to hear that you found our manuscript to be insightful and informative. We have taken your suggestion into account and have added a table on current and potential FD therapies that includes information on type of study, efficacy, and possible complications. Please see Table 2.Round 2
Reviewer 1 Report
The quality of the manuscript has improved substantially after the revision.